# Abdominal Lymphadenopathies: Lymphoma, Brucellosis or Tuberculosis? Multidisciplinary Approach—Case Report and Review of the Literature

**DOI:** 10.3390/medicina59020293

**Published:** 2023-02-04

**Authors:** Antonio Mirijello, Noemi Ritrovato, Angelo D’Agruma, Angela de Matthaeis, Luca Pazienza, Paola Parente, Dario Pio Cassano, Annalucia Biancofiore, Angelo Ambrosio, Illuminato Carosi, Ettore Serricchio, Paolo Graziano, Francesca Bazzocchi, Pamela Piscitelli, Salvatore De Cosmo

**Affiliations:** 1Unit of Internal Medicine, Department of Medical Sciences, Fondazione IRCCS Casa Sollievo della Sofferenza, 71013 San Giovanni Rotondo, Italy; 2Unit of Radiology, Fondazione IRCCS Casa Sollievo della Sofferenza, 71013 San Giovanni Rotondo, Italy; 3Unit of Pathology, Fondazione IRCCS Casa Sollievo della Sofferenza, 71013 San Giovanni Rotondo, Italy; 4Unit of Abdominal Surgery, Department of Surgical Sciences, Fondazione IRCCS Casa Sollievo della Sofferenza, 71013 San Giovanni Rotondo, Italy; 5Unit of Pharmacy, Department of Pharmaceuticals, Fondazione IRCCS Casa Sollievo della Sofferenza, 71013 San Giovanni Rotondo, Italy

**Keywords:** abdominal pain, fever, lymphadenopathies, brucellosis, tuberculosis

## Abstract

Abdominal pain represents a frequent symptom for referral to emergency departments and/or internal medicine outpatient setting. Similarly, fever, fatigue and weight loss are non-specific manifestations of disease. The present case describes the diagnostic process in a patient with abdominal pain and a palpable abdominal mass. Abdominal ultrasonography confirmed the presence of a mass in the mesogastrium. Computed Tomography (CT) and Magnetic Resonance Imaging (MRI) scans oriented toward calcific lymphadenopathies with increased metabolism in the positron emission tomography–computed tomography (PET-CT) scan. Laboratory examinations were inconclusive, although serology for Brucella and the Quantiferon test were positive. After multidisciplinary discussion, the patient underwent surgical excision of the abdominal mass. Histological examination excluded malignancies and oriented toward brucellosis in a patient with latent tuberculosis. The patient was treated with rifampin 600 mg qd and doxycycline 100 mg bid for 6 weeks with resolution of the symptoms. In addition, rifampin was continued for a total of 6 months in order to treat latent tuberculosis. This case underlines the need for a multidisciplinary approach in the diagnostic approach to abdominal lymphadenopathies.

## 1. Introduction

Abdominal pain represents one of the main reasons for referral to emergency department (ED) [1] and to internal medicine outpatient setting [2]. Given that the etiology of abdominal symptoms can be due to a wide range of diseases, a systematic approach is pivotal to reach the correct diagnosis. In fact, it has been shown that clinical history and physical examination alone are able to discriminate between organic and non-organic causes of abdominal pain in 79% of patients [1].

Enlarged lymph nodes can be a manifestation of infective, autoimmune and neoplastic diseases [3]. Similarly, fever, fatigue and weight loss are non-specific symptoms. Thus, clinicians should be guided by semiotics in the preliminary evaluation of patients in order to decide on the optimal diagnostic workup to reach the diagnosis [4].

Brucellosis is an old disease, widely distributed across the globe. Its incidence has re-emerged in the last decade [5]. Brucellae are Gram-negative, aerobic, facultative intracellular rods or coccobacilli [6], with 12 recognized species, 4 of which cause disease in humans. Clinical manifestations can range from asymptomatic presentations to multiorgan involvement [7]. The most common symptoms are fever, night sweats, chills, headaches and abdominal pain. However, although rare, isolated lymphadenopathy is a possible presentation.

Even tuberculosis (TB) represents a re-emerging disease [8]. Similarly to Brucellosis, its clinical manifestations can be non-specific, including fever, night sweats, weight loss or hemoptysis [9]. Isolated abdominal lymphadenopathies represent about 1% of TB cases [10].

The present report describes the diagnostic approach to a patient with abdominal pain and a palpable abdominal mass; an integration of clinical history together with laboratory, radiology and histology was needed to reach a diagnosis and to establish treatment.

## 2. Case Description

A 65-year-old Italian woman with significant past medical history of chronic hydrocephalus, Addison’s disease, osteoporosis and depression was referred to our outpatient clinic because of abdominal pain. The pain was prevalently nocturnal, requiring a painkiller (ketoprofen) almost every night. She reported regular hive and no association of symptoms with a meal.

At physical examination, the abdomen was soft. A non-tender but painful mass was palpable in the hypogastrium. No other significant signs were found. Abdominal ultrasound (US) scan revealed, in the mesogastrium, the presence of a hypoechoic dishomogeneous round mass, with internal calcifications, surrounded by thickened bowel loops. The patient was admitted to our inpatient unit for further examination.

Laboratory tests showed normocytic anemia, elevated inflammatory indices with a trend toward leukopenia. The other laboratory tests were normal (Table 1).

Colonoscopy and gynecological examination with endocavitary pelvic US scan were negative.

Due to iodinated contrast agent allergy, the patient underwent abdominal contrast-enhanced magnetic resonance imaging (CE-MRI) scan complete with computed tomography (CT) scans documenting, in the hypogastrium, the presence of voluminous multilobular lymphadenopathies (about 45 mm of maximum diameter) with calcific nucleus surrounding the mesenteric vessels and infiltrating the mesenteric fat (Figure 1).

A subsequent positron emission tomography–computed tomography (PET-CT) scan revealed the presence of diffuse and pathological accumulation of the radiotracer (max SUV 13.6) in correspondence with the lymphatic nodules seen in magnetic resonance imaging (MRI) (Figure 2).

These results—abdominal lymphadenopathies of unknown origin—required further diagnostic evaluations.

First of all, upon deeper collection of medical history, the patient reported a 7-day fever associated with chills and night sweats, occurring about 3 months before the appearance of abdominal symptoms. This symptomatology was self-limiting.

Serologies for the Epstein–Barr virus (EBV) and cytomegalovirus (CMV) were consistent with previous infection, while hepatitis B virus (HBV) and hepatitis C virus (HCV) antibodies were negative (Table 2A).

Quantiferon-TB Gold Plus emerged as positive, in absence of pulmonary lesions in the PET-CT scan; the Wright test for brucellosis was positive (1/400) (Table 2B).

After interdisciplinary discussion, the patient was scheduled for surgical excision of the abdominal neoformation. The surgical procedure is summarized as follows:

“Pneumoperitoneum with open technique in the supraumbilical site. Introduction of 2 trocars. Exploration of the abdominal cavity reveals a voluminous nodule at the level of the meso-ileum; multiple ileal loops and a section of the transverse colon are attracted to the nodule. Considering the anatomical-surgical picture, laparotomic conversion is necessary. Supra-sub-umbilical midline incision. Release of the afore-mentioned intestinal loops from the neoformation and removal of the nodule which is sent for extemporaneous histological examination (non-neoplastic process, possibly infectious)…”.

The definitive histological examination is reported below:

“Macroscopically, the surgical specimen consisted of solid nodular fragment, whitish in color, 4 × 3.5 × 3 cm in size, homogeneous in cut surface, partially calcified. Microscopically, Hematoxylin and Eosin stained sections documented extensive fibrosis with dystrophic calcifications and foci of lymphomonocytic, plasma cell and granulocytic infiltrate, mixed with areas of necrosis. To investigate the presence of fungi and/or mycobacteria, histochemistry was performed with Periodic Acid-Schiff (PAS), PAS-D (Periodic Acid-Schiff with diastase), Grocott’s and Ziehl- Neelsen stains in seriate sections. No fungi and mycobacteria were detected. A descriptive final diagnosis of fibro-hyaline nodule with mixed inflammatory infiltrate and suppurative necrosis was made, advising to carry out culture tests and integration with clinical-serological data” (Figure 3).

These findings, coupled with the high count of Brucella antibodies, excluded malignancies (i.e., lymphoma) and oriented toward a subacute/chronic infectious disease. With this in mind, real-time polymerase chain reaction (RT-PCR) to determine Brucella deoxyribonucleic acid (DNA) or mycobacteria was required. Unfortunately, these procedures were not possible after paraffin fixation.

A specific re-evaluation of clinical history was necessary; the patient confirmed the consumption of unpasteurized milk about 15 days before the appearance of fever.

Clinical, microbiological and histological data were consistent with mesenteric lymphadenopathies due to Brucella infection. For this reason, treatment with rifampin 600 mg daily and doxycycline 100 mg twice a day for 6 weeks was prescribed. Considering the positivity of Quantiferon, rifampin was continued for a total of 6 months in order to also treat latent tuberculosis (TB). After 4 weeks of treatment, blood exams and the abdominal US scan were normal. At the end of treatment course, the Wright serology became negative, and Quantiferon TB was indeterminate. The abdominal CT scan performed three months after the end of treatment was within normality.

## 3. Discussion

The present case describes the finding of mesenteric lymphadenopathies in a patient with abdominal pain and a previous history of fever, night sweats and consumption of unpasteurized milk. First-line lab tests were non-specific, while radiological exams could orient toward a suspicion of lymphoma. However, the Wright serology and Quantiferon were positive, and histology led to the diagnosis of brucellosis in a patient with latent TB.

According to the literature, the most common causes of mesenteric lymphadenopathies are inflammatory, infectious and neoplastic diseases [11]. In the present case, the inflammatory state shown in first-line lab examinations was not specific. The common causes of infectious diseases involving lymph nodes were consistent with brucellosis and TB (Table 2B). Finally, histology excluded malignancies (e.g., lymphoma and cancer).

Brucellosis is a common zoonosis, affecting half a million people annually [12]. The most common route of infection is consumption of unpasteurized milk or milk products. Four species of Brucella have been reported to cause human disease: *Brucella melitensis* (being the most common), *Brucella abortus*, *Brucella suis* and *Brucella canis* [5,13].

Brucellosis has a wide spectrum of clinical manifestations, ranging from asymptomatic presentations to multiorgan involvement [7]. In addition, Brucellosis can mimic any disease, making its diagnosis difficult [14]. The most common symptoms are fever, night sweats, chills, headaches and joint pain, although any organ can be affected. Moreover, brucellosis can manifest with isolated abdominal lymphadenopathies [15]. This last presentation is more common among children than adults [16].

Table 3 summarizes, in chronological order, the clinical and diagnostic features of case reports describing isolated abdominal lymphadenopathies associated with brucellosis, as reported in the English literature. Although rare, this presentation has been reported to account for 2.4–19% of cases [7,15,17].

In a retrospective evaluation of 1028 brucellosis cases, lymphadenopathy was present in 2.4% of cases [15]. Of these, 15 patients were affected by acute brucellosis, 6 patients by subacute and 4 by the chronic form of disease [15]. In a different series of 72 patients, the prevalence of lymphadenopathies was 6.9% [7]. Descriptions coming from endemic areas report a higher prevalence of lymphadenopathies. For example, in a systematic review and meta-analysis of epidemiology and clinical manifestations of brucellosis in China, lymphadenectases were present in 19% of cases [17]. This percentage considers all lymph node stations, but cases with isolated involvement of abdominal lymph nodes are even more rare [12,18,19,20].

The English literature reports a few cases similar to the present one. In fact, Massoud et al. [12] and Jayakumar et al. [18] reported cases of patients suffering abdominal pain with clinical data consistent with lymphoma. In both cases, histology concluded reactive lymphadenitis caused by B. melitensis [12,18]. Rodrigues Dos Santos et al. [20] even described a case of brucellosis with enlarged abdominal lymph nodes and possible ileal involvement. Acute abdominal pain itself can be read as a symptom of brucellosis, as suggested by Göke et al. in their case [19]. In this regard, US scan seems to be promising for supporting the diagnosis. In fact, in a Turkish study evaluating the role of abdominal sonography in a series of 251 cases of Brucellosis, Pourbagher et al. found that the frequency of enlarged abdominal (periportal) lymph nodes in patients with acute, subacute and chronic disease was 9.2% [21]. Considering these data, careful clinical history analysis and differential diagnosis are necessary for the correct approach.

It is noteworthy to underline that the immune response to Brucella infection involves both innate and adaptive immunity [22]. Macrophages are active participants in this process because B. melitensis can survive to their phagocytosis, evading the innate immunity [23]. This leads to lymphocytes recruitment and interferon (IFN) gamma production, finally contributing to lymph node enlargement. All these pathophysiological mechanisms are shared with TB. In particular, the molecular strategies to escape to macrophage killing (e.g., inhibition of phagosome–lysosome fusion) are similar between these two diseases [24,25]. Interestingly, Masoudian et al. [26] found the expression of similar genes in THP-1 macrophages of both diseases [26]. Finally, even several clinical features are common in TB and Brucellosis (e.g., fever, lymphadenopathy, hepato-splenomegaly) [27]. In this regard, vertebral osteomyelitis and posterior abscess can be found as extrapulmonary manifestations of TB (Pott’s disease) or spinal localization of Brucella (pseudo-Pott’s disease). As already stated, fever is a non-specific symptom. However, its feature can be seen as a potential tool for distinction. In fact, TB generally causes low-grade sub-continuous fever, whereas Brucella spp. are known to provoke intermittent fever [28]. Finally, both TB and Brucella spp. can cause hepato-splenomegaly (mainly Brucella spp.) and central nervous system (CNS) infections [29]. The sharing of common cytokines, immunity cells, molecular pathways can justify the presentation with granuloma in both pathological settings. However, TB typically produces caseous necrosis, while Brucella spp. does not [30]. However, differential diagnosis is not easy and requires laboratory and radiology.

## 4. Conclusions

Fever and abdominal pain represent non-specific symptoms, requiring a complete medical workup. Both Brucellosis and TB can manifest with isolated lymph node enlargement. The integration of medical history, laboratory and instrumental examinations is required to reach the correct diagnosis. The criterion ex juvantibus could be adopted when a definite diagnosis cannot be made.

## Figures and Tables

**Figure 1 medicina-59-00293-f001:**
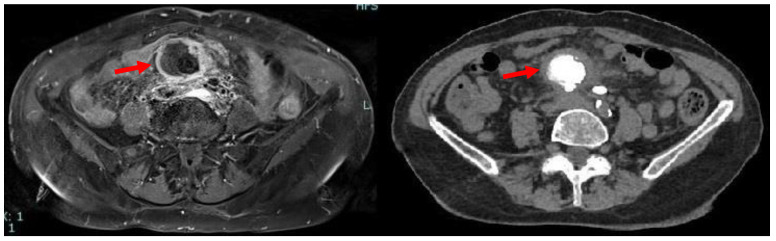
Axial MRI (**left**) and CT (**right**) scan showing the presence, in mesogastrium, of a nodular mass (red arrows) with contrast enhancement at MRI and internal calcifications at CT.

**Figure 2 medicina-59-00293-f002:**
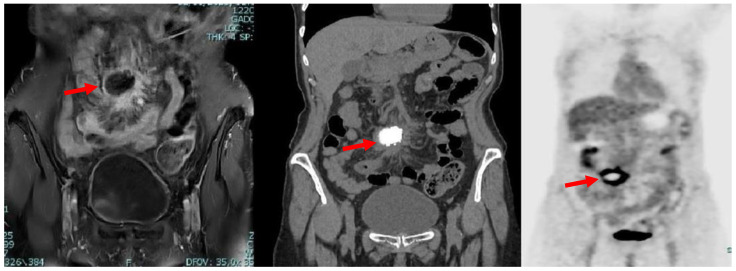
Coronal MRI (**left**), CT (**middle**) and PET (**right**) scans showing the presence, in mesogastrium, of a nodular mass (red arrows) with contrast enhancement at MRI and internal calcifications at CT, with thickened mesenterial fat and significant uptake of the radiotracer at PET-CT.

**Figure 3 medicina-59-00293-f003:**
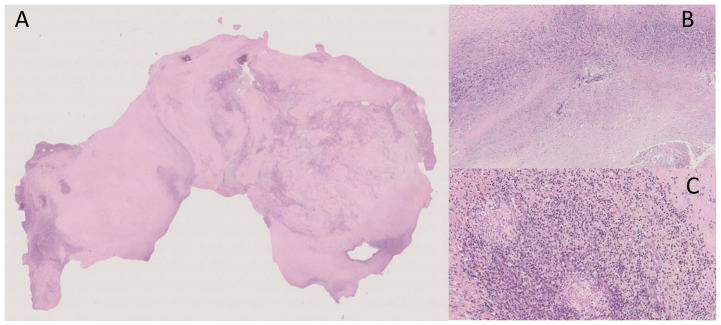
(**A**) Extensive fibrosis and dystrophic calcification (Hematoxylin and Eosin, 0.5×); (**B**) Suppurative necrosis (Hematoxylin and Eosin, 5×); (**C**) Lymphomonocytic, plasma cell and granulocytic infiltrate (Hematoxylin and Eosin, 20×).

**Table 1 medicina-59-00293-t001:** Results of main laboratory examinations performed; altered values in bold. (Abbreviations: MCV = mean corpuscular volume; PLT = platelet count; WBC = white blood cell count; ESR = erythrocyte sedimentation rate; PCR = polymerase chain reaction).

	Results	Normal Values
Hemoglobin (g/dL)	**10.8**	14.00–18.00
MCV (fl)	85	77.00–98.00
PLT (U/mm³)	261.000	130.00–400.00
WBC (U/mm³)	4840	4.30–10.80
ESR (mm/h)	**42**	2.0–15.0
PCR (mg/dL)	**2.39**	<0.30
Glycemia (mg/dL)	67	60–100
Total blood proteins (g/dL)	5.8	6.40–8.20
Albumin/serum (g/dL)	**3.1**	3.50–5.50
Sideremia (mcg/dL)	**41**	65.00–175.00
Ferritin (ng/mL)	**36**	26.0–388.0
Transferrin (mg/dL)	178	200–360
Transferrin saturation levels (%)	**18.4**	20–50
Albumin (%)	**55.1**	55.80–66.10
Alfa-1-globulin (%)	**6.6%**	2.90–4.90
Alfa-2-globulin (%)	**13.6%**	7.10–11.80
Beta-1-globulin (%)	**6.3%**	4.70–7.20
Beta-2-globulin (%)	**5.3%**	3.20–6.50
Gamma-globulin (%)	**13.1%**	11.10–18.50

**Table 2 medicina-59-00293-t002:** (**A**) Antibody profile of the main infectious diseases (Abbreviations: EBV: Epstein–Barr Virus, CMV: Cytomegalovirus, IgM: Immunoglobulin M, IgG: Immunoglobulin G). (**B**) Antibody profile of the main infectious diseases.

(**A**)		
	**IgM**	**IgG**
Toxoplasmosis	-	+
Rubella	-	+
CMV	-	+
EBV	-	+
Herpes Simplex	-	-
(**B**)		
	**Result (t1)**	**Result (t2)**
Salmonella Typhi	-	-
Parathyfoid B	-	-
Brucella Melitensis	+	+
Quantiferon TB-Gold Plus	+	INDETERMINATE

**Table 3 medicina-59-00293-t003:** Cases of isolated abdominal lymphadenopathies associated with brucellosis reported in the literature (chronological order). (Abbreviations: CRP: C-reactive protein; CT: Computed tomography; ELISA: Enzyme Linked ImmunoSorbent Assay; ESR: Erythrocyte sedimentation rate; IgM: Immunoglobulin M; IgG: Immunoglobulin G; WBC: White blood cell; M: Male; F: Female; n.a.: Not available).

	Authors	Age	Gender	Symptoms/Signs	Lab Exams	Radiology/Biopsy	Other
1	Jayakumar et al. [18]	19	M	Abdominal pain, fever, vomit, abdominal rigidity, tenderness in right iliac fossa	Slightly elevated WBC	Lymph node laparotomy	Blood culture +
2	Göke et al. [19]	34	F	Septic fever, sweats, abdominal pain, arthralgia, hepato-splenomegaly	Pancytopenia	n.a.	n.a.
3	Massoud et al. [12]	52	M	Fever, chills, night sweats, abdominal pain	Normal WBC	CT scan Lymph node biopsy	Wright test + (1/1280) Blood culture +
4	Rodrigues Dos Santos et al. [20]	68	F	Night sweats, anorexia, colicky abdominal pain, fever, diarrhea	Leukopenia, anemia, slightly elevated CRP and ESR	Abdominal US Chest X-ray CT scan Lymph node biopsy	Blood culture + Rose-Bengal test + ELISA IgM + IgG −

## Data Availability

Data supporting reported results may be provided on reasonable request.

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
