# Peer review of "Abdominal Lymphadenopathies: Lymphoma, Brucellosis or Tuberculosis? Multidisciplinary Approach—Case Report and Review of the Literature"

_medicina, 2023, doi:10.3390/medicina59020293_

Round 1

Reviewer 1 Report

The current case report has addressed the diagnostic approach to a patient with abdominal pain and a palpable abdominal mass. This case report shows that Brucellosis and Tuberculosis can present with enlargement of isolated lymph nodes. This manuscript is poorly written although it is an rare case.  The quality of the manuscript is not sufficient for publication.  I would like to offer the following minor points for consideration by the authors towards the improvement of the manuscript:

1) There are a number of grammatical/spelling errors which need correcting. Some examples include:

P1-Line 38 “weigh loss”

2- Previous published case reports on clinicopathologic features of  mesenteric lymphadenopathies mimicking lymphoma or mesenteric lymphadenopathies in patients with Brucellosis can be tabulated.

3- In the title and abstract, it can be stated that the case is related to Brucellosis.

4- The introduction and discussion section are scarce and they do not sufficiently highlight the originality and the significant contributions of the study. Therefore, they need to be reformulated.

5-  Please mention A, B, and C in Figure 3.

6- Only 20 % of references are from the last 5 years. The inclusion of more recent results would be highly appreciated by the general reader interested in this topic.

Author Response

We thank the Reviewer for the suggestions aiming at improving the quality and readability of our manuscript.

All the raised comments have been addressed in the text. In particular:

  • COMMENT: There are a number of grammatical/spelling errors which need correcting. Some examples include: P1-Line 38 “weigh loss”
    1. REPLY: English Language has been entirely revised throughout the text (changes in red) .
  • Previous published case reports on clinicopathologic features of mesenteric lymphadenopathies mimicking lymphoma or mesenteric lymphadenopathies in patients with Brucellosis can be tabulated.
    1. REPLY: As suggested by the Reviewer, previous published case reports describing mesenteric lymphadenopathies associated with Brucella infection have been reported in TABLE 3, while most significant case series and reviews have been discussed in the text.
  • In the title and abstract, it can be stated that the case is related to Brucellosis.
    1. REPLY: We appreciated this suggestion. The title has now been changed into “Abdominal lymphadenopathies: lymphoma, brucellosis or tu-berculosis? Multidisciplinary approach. Case report and review of the literature” in order to include both brucellosis and TB.
  • The introduction and discussion section are scarce and they do not sufficiently highlight the originality and the significant contributions of the study. Therefore, they need to be reformulated.
    1. REPLY: thank you for this important suggestion. Introduction has now been rewritten in order to include information on both brucellosis and TB. Discussion section has been expanded.
  • Please mention A, B, and C in Figure 3.
    1. REPLY: amended
  • Only 20 % of references are from the last 5 years. The inclusion of more recent results would be highly appreciated by the general reader interested in this topic
    1. REPLY: We have added recent references to the text See ref. n. 5, 6, 8, 9, 10, 13, 16, 18, 21.

Reviewer 2 Report

This is a case report of Brucellosis Lymphadenopathy diagnosed by Clinical, microbiological and histological investigations.

The title of the case report should say something about “Brucella infection”.

Abstract and Introduction:

In your abstract and introduction, can you write about the target disease rather than general non-specific symptoms? Abdominal pain is not the subject of your case report.

Line 38: Spelling mistakes: asthenia (Myasthenia), weigh (weight)

Line 39: Spelling mistakes: semeiotics (semiotics)

Figure 1, 2: Can you use arrows to refer to the mass?

How did you make the decision for surgical resection? Is there an interdisciplinary consensus?

Line 97: Can you describe in detail the surgical procedure and findings? (Some of the co-authors are from the unit of abdominal surgery)

Conclusions:

“Fever and abdominal pain represent non-specific signs of disease, requiring a complete medical workup.” What disease?

Your case report was about Brucellosis. Why do you write about tuberculosis in your conclusions?

Author Response

REVIEWER #2

We sincerely appreciated the Reviewer’s suggestions to our manuscript. All the raised comments have been addressed in the text. In particular:

  • The title of the case report should say something about “Brucella infection”.
    1. REPLY We appreciated this suggestion. As also suggested by the other Reviewer, the title has now been changed into “Abdominal lymphadenopathies: lymphoma, brucellosis or tu-berculosis? Multidisciplinary approach. Case report and review of the literature”
  • In your abstract and introduction, can you write about the target disease rather than general non-specific symptoms? Abdominal pain is not the subject of your case report.
    1. REPLY: Thank you for this observation. Abstract has been rewritten.
  • Line 38: Spelling mistakes:
    1. REPLY asthenia was replaced with fatigue, weigh with weight
  • Line 39: Spelling mistakes:
    1. REPLY: semeiotics à semiotics
  • Figure 1, 2: Can you use arrows to refer to the mass?
    1. REPLY: Arrows have been added
  • How did you make the decision for surgical resection? Is there an interdisciplinary consensus?
    1. REPLY: The decision for surgery was multidisciplinary. We specified it in the text. Thank you.
  • Line 97: Can you describe in detail the surgical procedure and findings? (Some of the co-authors are from the unit of abdominal surgery)
    1. REPLY: Surgical description was added
  • Conclusions: “Fever and abdominal pain represent non-specific signs of disease, requiring a complete medical workup.” What disease?
    1. REPLY: Thank you for this comment. We have modified “signs of disease” in “symptoms”.
  • Your case report was about Brucellosis. Why do you write about tuberculosis in your conclusions?
    1. REPLY: Thank you for this observation. Given that, in this specific case, TB was a possible differential diagnosis due to isolated abdominal lymphadenopathies, we decided to expand introduction and discussion sections on TB.

Round 2

Reviewer 1 Report

I am satisfied that the authors have addressed all of my previous concerns about the article. It is now much improved and I feel that it is now suitable for publication.

Reviewer 2 Report

The authors responded adequately to the comments.